# Effects of Silage-Based Diets and Cattle Efficiency Type on Performance, Profitability, and Predicted CH$_4$ Emission of Backgrounding Steers

**Mohammad Khakbazan** [1,*], **Hushton C. Block** [2], **John Huang** [1], **Jeff J. Colyn** [2], **Vern S. Baron** [2], **John A. Basarab** [3], **Changxi Li** [2,3] and **Chinyere Ekine-Dzivenu** [3]

[1] Agriculture and Agri-Food Canada (AAFC), Brandon Research and Development Centre, 2701 Grand Valley Rd, Brandon, MB R7A 5Y3, Canada; john.huang@agr.gc.ca
[2] AAFC, Lacombe Research and Development Centre, 6000 C & E Trail, Lacombe, AB T4L 1W1, Canada; hushton.block@agr.gc.ca (H.C.B.); jeff.colyn@agr.gc.ca (J.J.C.); vern.baron@agr.gc.ca (V.S.B.); changxi.li@agr.gc.ca (C.L.)
[3] Department of Agricultural, Food and Nutritional Science, University of Alberta, 16 St & 85 Ave, Edmonton, AB T6G 2P5, Canada; jbasarab@ualberta.ca (J.A.B.); c.ekine@cgiar.org (C.E.-D.)
[*] Correspondence: mohammad.khakbazan@agr.gc.ca

**Abstract:** Feed costs are the largest expense in commercial beef production. Increasing cattle (*Bos taurus*) feed efficiency should reduce feed costs and increase beef profitability. This study used data from two years of a backgrounding trial conducted in Lacombe, Alberta, Canada. The evaluation looked at economic and predicted CH$_4$ emission impacts of diet quality and cattle efficiency type in backgrounding systems. The hypothesis was that diet quality from use of barley (*Hordeum vulgare* c.v. Canmore) or triticale (*x Triticosecale* c.v. Bunker) silage-based diets and cattle efficiency type defined by residual feed intake would interact to affect profitability and CH$_4$ emissions. Effects of diet and cattle efficiency type on profitability and CO$_2$e emissions were assessed using statistical and stochastic risk simulation. The profitability of beef backgrounding was affected by cattle efficiency type and diet quality with higher quality barley silage also lowering CO$_2$e emissions. The difference in certainty equivalent (CAD~30 steer$^{-1}$) of efficient steers on barley silage and inefficient steers on barley silage or efficient or inefficient steers on triticale silage supports a beef backgrounding producer focus on diet quality and cattle efficiency type. This study did not address potential agronomic differences, including yield, which could provide nuance to forage choice.

**Keywords:** cattle performance; net revenue; risk analysis

## 1. Introduction

Feed costs are the single largest expense in most commercial beef production [1,2], and feeding and management through the winter comprise up to two-thirds of the total cost of primary beef production in Canada [3]. Increasing cattle feed efficiency should increase the profitability and economic sustainability of beef production.

There is individual animal variation in feed efficiency. Residual feed intake (RFI) [4] is the difference between actual feed intake and expected intake given the observed weight and gain [5]. It is a measure of feed efficiency that is independent of body size, production, and growth traits [6,7]. This increases its utility for comparisons across production levels and production phases for the selection of more efficient cattle types [7,8]. The moderate repeatability [5] and moderate heritability [7,9,10] of RFI should allow for the selection of more efficient cattle [11,12].

Selection for RFI has been shown to reduce feed costs [13,14] and increase economic benefits [7,15,16]. Selection for low-RFI (efficient) cattle can also have a variety of environmental benefits, including reduced methane emissions [7,17], reduced manure production [7], and smaller farm area requirements [5].

The measurement of individual animal intake for accurate estimation of RFI usually requires access to high-tech feeding facilities [18]. Testing for RFI is also expensive and slow to return economic benefits [7,12]. The benefits of selecting for feed efficiency in cattle have not yet been recognized by many producers, further limiting the widespread adoption of RFI selection [19]. A survey of 269 commercial cow-calf producers in the United States found that only one-third of surveyed producers correctly identified the definition of feed efficiency, and only 15% of producers had any knowledge of residual feed intake [20]. Consequently, the measurement of cattle RFI has thus far been limited to researchers and progressive seed stock producers.

The increasing access to RFI measurement has generated interest in the use of genomic technologies to predict genomic expected progeny differences (gEPD) and breeding values [21–24]. Genomic selection of RFI allows for the moderately accurate selection of cattle early in life [25], providing producers with timely access to feed efficiency data that can be used for the selection of efficient cattle.

The responsiveness of cattle to different quality diets is relatively well understood [26], allowing joint consideration of yield and quality with crop selection models [27]. Despite the expected comparable economic value of silage from Canmore barley and Bunker triticale, the compositional difference is expected to result in a quality difference [27]. In addition to the independent economic and environmental benefits from improving cattle efficiency through managing nutrition and selecting for efficient cattle, there is anecdotal evidence that an interaction between cattle feed efficiency and production environment exists. Unpublished research from the Lacombe Research Centre found that low-RFI beef cows gained more body fat and body weight (BW) than high-RFI cows when both groups swath grazed forages under extensive Canadian winter conditions. This suggests that efficient animals may be more adaptable and less susceptible to stress than inefficient animals [5]. The objective of this study was to evaluate the economic effects of barley or triticale silage-based diets and cattle efficiency type for backgrounding steers. The choice of silages was based on the crop selection model mentioned [27]. There is no known direct comparison of these two silages in other cattle feeding studies. Additionally, stochastic simulation, which allows cost, revenue, and production factors to be analyzed as statistical distributions rather than as point estimates [28–31], was used to assess probable cattle producer preferences. Finally, the performance of animals in terms of $CH_4$ emissions (in $CO_2$ equivalent ($CO_2$e)) under different diet treatments and cattle efficiency types was investigated. The hypothesis was that differences in diet quality from use of barley or triticale silage-based diets would interact with beef cattle efficiency type and affect backgrounding profitability and $CO_2$e emissions.

## 2. Materials and Methods

### 2.1. Experimental Design

The steers used in this two-year study were maintained at the Lacombe Research and Development Centre, Agriculture and Agri-Food Canada, Lacombe, Alberta. All dietary treatments and experimental procedures related to animal use and care were approved on 18 April 2016 by the Lacombe Research and Development Centre Animal Care Committee (LRC Study Plan No. 201602) prior to any animal-related research activity with animal care in accordance with the guidelines established by the Canadian Council on Animal Care [32]. An amendment to allow additional sampling was approved on 18 January 2017 with interim and final animal use reports accepted on 18 April 2017 and 16 April 2018. The study complies with the ARRIVE guidelines.

Calves were born between the beginning of March and mid-May in 2016 (year 1) and 2017 (year 2). The start of trial age and weight ($\mu \pm$ SD) of steers was $198.5 \pm 15.4$ d and $288.4 \pm 30.5$ kg for year 1 and $229.0 \pm 13.3$ d and $280.6 \pm 41.8$ kg for year 2. The experimental design for the animal evaluation trials was a four (cattle efficiency type) by two (diet quality) factorial treatment structure with two replicates (pens of 8 steers) per treatment combination repeated over two years in a randomized complete block design. Prior to the backgrounding

trials, 128 steer calves were tissue sampled for genotyping on Illumina Bovine SNP50 Beadchip and prediction of RFI. The prediction of RFI was conducted with the Genomic Best Linear Unbiased Prediction (GBLUP) method as described previously [23,33], using Canadian beef cattle with both genotypes and RFI as a reference population (N = 4583). Genotyping before weaning allowed the sorting of calves into cattle efficiency type groups before an RFI test could be completed. In each of the two 112-day trials, which ran from October 2016 to January 2017 and November 2017 to February 2018, the calves were split into quartiles based on 50 k SNP predicted RFI, stratified by weight, and randomized to pens. Within each cattle efficiency quartile (4 pens), each pen was randomized to a barley (*Hordeum vulgare* c.v. Canmore) silage- or triticale (*x Triticosecale* c.v. Bunker) silage-based diet (Table 1).

**Table 1.** Means and standard deviation (±SD) for live steer weight, predicted and observed residual feed intake (RFI), and steer average daily gain (ADG) for diet treatments over the two-year study.

| Experimental Year | Diet | Predicted RFI Quartile | Live Weight at Start of Experiment (kg steer$^{-1}$) | Live Weight at End of Experiment (kg steer$^{-1}$) | Predicted RFI | Observed RFI | ADG (kg Steer$^{-1}$ Day$^{-1}$) |
|---|---|---|---|---|---|---|---|
| 1 | Barley silage | 1 | 283 ± 2.27 | 392 ± 5.32 | −0.131 ± 0.002 | −0.124 ± 0.139 | 0.977 ± 0.03 |
| 1 | Barley silage | 2 | 288 ± 2.63 | 387 ± 7.70 | −0.024 ± 0.002 | 0.140 ± 0.081 | 0.885 ± 0.05 |
| 1 | Barley silage | 3 | 291 ± 3.62 | 389 ± 16.22 | 0.065 ± 0.002 | −0.024 ± 0.019 | 0.876 ± 0.11 |
| 1 | Barley silage | 4 | 285 ± 3.02 | 377 ± 5.53 | 0.158 ± 0.001 | 0.008 ± 0.408 | 0.821 ± 0.02 |
| 1 | Triticale silage | 1 | 282 ± 4.62 | 372 ± 2.08 | −0.133 ± 0.002 | −0.051 ± 0.438 | 0.803 ± 0.02 |
| 1 | Triticale silage | 2 | 290 ± 3.33 | 387 ± 11.8 | −0.024 ± 0.000 | −0.075 ± 0.060 | 0.869 ± 0.08 |
| 1 | Triticale silage | 3 | 284 ± 14.99 | 387 ± 13.66 | 0.068 ± 0.008 | 0.249 ± 0.006 | 0.921 ± 0.01 |
| 1 | Triticale silage | 4 | 290 ± 5.83 | 386 ± 4.10 | 0.160 ± 0.005 | −0.123 ± 0.022 | 0.858 ± 0.09 |
| 2 | Barley silage | 1 | 278 ± 3.16 | 399 ± 4.26 | −0.115 ± 0.003 | 0.140 ± 0.472 | 1.081 ± 0.01 |
| 2 | Barley silage | 2 | 274 ± 5.34 | 391 ± 9.08 | −0.019 ± 0.000 | −0.011 ± 0.209 | 1.039 ± 0.03 |
| 2 | Barley silage | 3 | 295 ± 1.89 | 409 ± 7.94 | 0.055 ± 0.000 | 0.144 ± 0.484 | 1.017 ± 0.09 |
| 2 | Barley silage | 4 | 278 ± 1.35 | 395 ± 9.46 | 0.137 ± 0.001 | −0.273 ± 0.148 | 1.037 ± 0.10 |
| 2 | Triticale silage | 1 | 278 ± 0.98 | 375 ± 10.36 | −0.117 ± 0.008 | 0.135 ± 0.341 | 0.868 ± 0.08 |
| 2 | Triticale silage | 2 | 271 ± 1.14 | 376 ± 3.47 | −0.020 ± 0.001 | −0.124 ± 0.173 | 0.941 ± 0.04 |
| 2 | Triticale silage | 3 | 294 ± 8.58 | 404 ± 0.65 | 0.053 ± 0.001 | −0.129 ± 0.163 | 0.980 ± 0.08 |
| 2 | Triticale silage | 4 | 277 ± 2.23 | 370 ± 5.02 | 0.141 ± 0.006 | 0.118 ± 0.498 | 0.832 ± 0.02 |

Note: average steer ages at start and end of experiment were 186 days and 298 days, respectively, for experimental year 1, and 206 days and 318 days, respectively, for experimental year 2. SD, standard deviation.

Based on wet chemistry analyses (Cumberland Valley Analytical Services, Waynesboro PA) for monthly composites of weekly feed samples collected throughout the study TDN content ((dry matter (DM) basis [34]) of barley silage was 62.8% in year 1 and 65.3% in year 2, whereas triticale silage was 59.9% in year 1 and 60.8% in year 2. Diets were formulated to minimize the expected feed cost of gain with steers expected to gain approximately 1 kg day$^{-1}$. The cDDGS-based supplement was included to address a metabolizable protein deficiency, while supplying additional Ca, monensin, and vitamins A, D, and E. Both diets in both years were approximately 75% silage and 25% corn (*Zea mais*) dried distillers' grains (cDDGS) on a basis. When wet chemistry analyses results were weighted by daily feed intakes for each pen, average final diet TDN content (DM basis [34]) of the barley silage diet was 71.1% in year 1 and 72.6% in year 2, whereas triticale silage was 68.8% in year 1 and 69.5% in year 2.

Animal performance data, including daily feed intake and monthly BW, were collected throughout the study. Steers were weighed in the morning before feeding on two consecutive days at the start and end of trial, and at 28-day intervals during the trials.

Dry matter intake (kg steer$^{-1}$ day$^{-1}$) was calculated from daily feed mixing and pen delivery records and weekly ingredient DM content. Weekly feed samples were composited monthly and analyzed to determine diet energy content. Feeding targeted ad libitum intake with slick bunk management to minimize feed waste. When orts occurred, they were weighed and a sub-sample dried to allow dry matter intake (DMI) and diet energy content correction. Feed efficiency was calculated from observed average daily gain (ADG) and DMI.

*2.2. Economic Analysis*

The revenue returns for the trial period were analyzed separately by using appropriate nine-year average (2010–2018) cattle prices at the beginning and the end of this phase [35]. Since steers were born in March and weaned in October, the feeder cattle price series for 136–408 kg weight classes used for the price distribution of steer calf purchase was the historical monthly price of October. Comparable January and February price series for feeder cattle weight classes within the 227–454 kg weight range were used for the price distribution of steer sales at the end of the feeding period. Industry price reporting is on the basis of an average price for specified weight classes. The average price for a given weight class varies seasonally. The prices in Table 2 represent the average price reported for cattle in each of the weight classes for the season in which cattle would have started on the trial and completed the trial. These prices were applied on an individual basis with consideration of the observed weight of the cattle at the start and end of the study. Prices for appropriate steer weight category were multiplied by the actual weight of steers, less a 3% shrinkage [36], to determine the initial and final price on a per steer basis. There was no seasonality adjustment for feeder prices as historical monthly prices for the month that calves were weaned or sold at the end of trial were used. The net revenue (NR) function (beef returns—variable costs) per steer for the different beef management treatments is expressed as follows:

$$NR = [P_E(W_F + ADG * Day)] - [P_F * W_F) + (P_B * W_B) + (P_T * W_T) + (P_{cDDGS} * W_{cDDGS}) + VC]$$

where $P_E$ is the price of steer at the end of trial; $W_F$ is the beginning weight of feeder; ADG is the average daily gain; Day is the number of days spent on feed; $P_F$ is the price of feeder steer at the beginning of trial; $P_B$ is the price of barley silage-based diet; $P_T$ is the price of triticale silage; $W_B$ is the weight of barley silage consumed; $W_T$ is the weight of triticale consumed, $P_{cDDGS}$ is the price of cDDGS, $W_{cDDGS}$ is the weight of cDDGS consumed, and VC is the other variable cost that was uniform across treatments. Variable costs include salt, bedding, yardage, manure removal, marketing, and trucking (Table 2). The NR was expressed in CAD steer$^{-1}$. Barley and triticale silage price discovery was a challenge as these ingredients are rarely traded. To address this challenge, the analysis was done under two scenarios: (A) the value of both the barley and triticale silages were estimated based on an industry standard practice of using the barley bushel price multiplied by 12.5 for a standard silage of 35% DM (CAD 56.6 tonne$^{-1}$) [37] and then adjusted for the actual DM of the silages used in the study [35]. (B) The value of barley and triticale silages were based on back calculation from known cattle prices and other input costs. Assuming all other prices and costs are known except the cost of silage, a breakeven point (NR = 0) was calculated and considered to represent the value of barley or triticale silages.

**Table 2.** Input and output prices. The average steer prices at beginning and end of trial were based on the Alberta Weekly Feeder Prices from 2010 to 2018 in CanFax.

| Item | Price |
|---|---|
| Steer at the beginning of trial at 136–181 kg (CAD kg$^{-1}$) | 5.08 |
| Steer at the beginning of trial at 181–227 kg (CAD kg$^{-1}$) | 4.85 |
| Steer at the beginning of trial at 227–272 kg (CAD kg$^{-1}$) | 4.51 |
| Steer at the beginning of trial at 272–318 kg (CAD kg$^{-1}$) | 4.10 |
| Steer at the beginning of trial at 318–363 kg (CAD kg$^{-1}$) | 3.76 |
| Steer at the beginning of trial at 363–408 kg (CAD kg$^{-1}$) | 3.55 |
| Steer at the ending of trial at 227–272 kg (CAD kg$^{-1}$) | 4.43 |
| Steer at the ending of trial at 272–318 kg (CAD kg$^{-1}$) | 4.17 |
| Steer at the ending of trial at 318–363 kg (CAD kg$^{-1}$) | 3.99 |
| Steer at the ending of trial at 363–408 kg (CAD kg$^{-1}$) | 3.81 |
| Steer at the ending of trial at 408–454 kg (CAD kg$^{-1}$) | 3.62 |

**Table 2.** *Cont.*

| Item | Price |
|------|-------|
| Silage (barley, triticale, 2010–2018, 65% moisture, CAD tonne$^{-1}$) | 56.57 |
| Adjusted barley silage price at 70.8% moisture for year 1 (CAD tonne$^{-1}$) | 47.22 |
| Adjusted triticale silage price at 58.6% moisture for year 1 (CAD tonne$^{-1}$) | 66.93 |
| Adjusted barley silage price at 65.1% moisture for year 2 (CAD tonne$^{-1}$) | 56.43 |
| Adjusted triticale silage price at 58.9% moisture for year 2 (CAD tonne$^{-1}$) | 66.30 |
| cDDGS [1], (CAD kg$^{-1}$) | 0.44 |
| Bedding straw (CAD tonne$^{-1}$) | 30.00 |
| Yardage (CAD day$^{-1}$) | 0.454 |
| Salt (CAD g$^{-1}$) | 0.0006 |
| Other [2] (CAD steer$^{-1}$) | 17.54 |

[1] cDDGS, corn dried distillers' grains. [2] Other: manure removal (CAD 0.02 steer$^{-1}$ day$^{-1}$), market commission (CAD 5 steer$^{-1}$), and steer trucking (CAD 0.0374 kg$^{-1}$).

*2.3. $CH_4$ Estimates*

The $CH_4$ emissions estimates for backgrounding steers were the median estimates obtained from the Nutrient Requirements of Beef Cattle model (NASEM) [26]. The intake and weighted diet composition observed for each pen of steers was entered into the model along with observed steer weights and body condition. The net energy for maintenance adjuster within the model was then used to get predicted ADG to equal observed ADG. The $CH_4$ emissions were the median of the model estimates. The discrepancy between observed and model predicted intake was noted and used in an additional simulation.

The NASEM [26] was also used to estimate steer performance under the hypothetical scenario of steers receiving a silage-only (i.e., no cDDGS-based supplement) diet. This scenario used the observed weighted average silage composition as the sole feed ingredient. Intake was based on model predicted intake with the same per cent adjustment as the discrepancy between observed and predicted intake noted previously for observed steer performance. The metabolizable energy limited gain was used to revise modeled steer final weight. Adjustments were iterative for each pen until predictions stabilized and expected intake, ADG, and methane emissions were recorded.

The $CH_4$ emissions were converted to $CO_2$e by multiplying by 25 [38] and expressed on a per steer per day, per kg of DMI, per kg of ADG, and a net revenue per kg of $CO_2$e emission basis.

*2.4. Statistical Analysis*

Observed pen average RFI was determined as the difference between DMI predicted from regression observed DMI on year, diet type, metabolic BW (BW$^{0.75}$), ADG, and final ultrasound backfat and observed DMI, as per Basarab et al. [16], except that regression used pen averages instead of individual animal data. Observed and predicted intakes were standardized to 10 MJ ME kg$^{-1}$ of diet DM, as appropriate for backgrounding or forage-based diets, before determination of RFI. Pen average observed RFI was determined separately for each year of the study, using all pens in each year.

The accuracy and precision of expected RFI values for pens of steers having randomized steers to pens based on genomic predicted RFI was assessed by comparing to observed pen average RFI by regression and concordance correlation coefficient determination [39,40].

All response data for this study were analyzed using the mixed models procedure of SAS [41,42]. The full statistical model was:

$$Y_{ijkl} = \mu + D_i + E_j + D_i \times E_j + E_j^2 + D_i \times E_j^2 + T_k + \varepsilon_{ijkl}$$

where: $\mu$ was the overall mean; $D_i$ was the effect (categorical) of diet; $E_j$ was the linear effect (quantitative) of observed RFI; $D_i \times E_j$ was interaction of diet with the linear effect of observed RFI; $E_j^2$ was the quadratic effect of observed RFI; $D_i \times E_j^2$ was the interaction of diet with the quadratic effect of observed RFI; $T_k$ was the random effect of year (blocking

factor); and $\varepsilon_{ijkl}$ was the error. Individual pen served as the experimental unit. An alpha of 0.05 was used for significance, and following a type I sums of squares analyses, a solution was fitted that included all relevant treatment effects and interactions. These analyses were carried out for gross cost and revenue, NR, silage breakeven value at both observed silage DM and a standardized 35% DM, and $CO_2$e for $CH_4$ emissions as predicted from observed performance. Analyses on DMI, ADG, and feed efficiency are included for context as major determinants of costs, revenue, and profitability.

The scenarios estimating $CH_4$ emissions with silage-only diets were analyzed for the effect of supplementation (actual silage-based diets vs. simulated silage-only diets) and silage (barley silage vs. triticale silage) as a 2 by 2 factorial using the mixed model procedure of SAS [41,42]. Pen was treated as a sub-plot factor for comparison between actual and simulated diets and year remained a random blocking factor. Effect of RFI was disregarded due to the observed impact on the same measures when considering only the actual diets, and the inability to determine an appropriate RFI for simulated scenarios.

*2.5. Risk Analysis—Stochastic Efficiency with Respect to a Function*

The Microsoft Excel add-in Simulation and Econometrics to Analyze Risk (SIMETAR), developed by Richardson et al. [43], was used to simulate feeder price at the beginning and end of trial, simulate ADG distributions, and calculate distributions of NR for steers. The multivariate empirical distributions derived from experimental ADG, was multiplied by a simulated price distribution derived from feeder historical prices (2010–2018) with SIMETAR software to calculate simulated NR for each treatment [44,45]. The simulated ADG was generated from the observed ADG for each of the treatments using 125 steers in 2016/2017 and 128 steers in 2017/2018. A two-sample Hotelling $T^2$ test was used to test for significant differences between the simulated data and the actual data [46]. SIMETAR was used to construct a cumulative probability distribution function (CDF) from simulated NR with probability ranging from 0.0 to 1.0 for barley silage-based diets vs. triticale silage-based diets and efficient (observed pen average RFI < 0) and inefficient (observed pen average RFI > 0) treatment combinations.

A constant relative risk-aversion function was used with stochastic efficiency with respect to a function (SERF) in this study to evaluate risk-efficient alternative treatments [43,47,48]. SERF identifies the most efficient alternative treatments for a range of risk preferences by ranking alternatives in terms of certainty equivalent (CE) [43]. The CE is a measure of a payoff that a decision maker (in this case backgrounding cattle producer) would have to receive to be indifferent between the certain payoff and a riskier alternative [47]. For a given level of absolute risk aversion coefficient (ARAC), the CE is calculated using the negative exponential equation defined by Pratt [49] and Hardaker et al. [47] as $r_a(w) = -u''(w)/u'(w)$, which represents the ratio of the second and first derivatives of the decision-maker's utility function, $u(w)$. The $r_a(w)$ is the risk aversion coefficient and $w$ is a measure of wealth. The negative exponential utility function assumes decision makers prefer less risk to more given the same expected return and they have constant absolute risk aversion (i.e., they view a risky strategy for a specific level of risk aversion the same without regard for their level of wealth). Across two or several alternatives, a higher CE, with the same level of ARAC is considered a best management alternative. The greater CE value with the same level of ARAC corresponds to a preferred alternative. The CE values were estimated for efficient steers (observed RFI less than zero) on barley silage-based diets, inefficient steers (observed RFI greater than zero) on barley silage-based diets, efficient steers (observed RFI less than zero) on triticale silage-based diets, and inefficient steers (observed RFI greater than zero) on triticale silage-based diets.

The CE values were calculated for the ARAC for an upper boundary of 4/average NR [47]. The ARAC ranged from 0 (risk neutral) to 0.005 (highly risk averse). There were 500 simulated NR values computed (data not shown). The simulated NR values were then used to determine the risk premium to evaluate the preferred strategies under risk.

### 3. Results

#### 3.1. Adequacy of RFI Predictions

With the predicted pen average RFI values used in assigning steers to pens based on RFI quartiles having no relationship ($p = 0.67$) to observed pen average RFI, deviation ($p < 0.05$) from the isopleth, and a low concordance correlation coefficient of $-0.054$ due to a low precision coefficient of $-0.077$, subsequent analyses are based on regression on observed pen average RFI to describe cattle efficiency type, instead of predicted RFI quartile. Pen of steers remained as the experimental unit.

#### 3.2. Economic Analyses Data

The unit prices for cattle, feed, bedding, yardage, and other costs used in this economic evaluation are presented in Table 2. These prices indicate that the price of steers declines by 0.17% per kg increase in weight for steer calves bought in October, and there is a 0.11% per kg increase in weight for steers sold in January or February. This compares to a multi-year industry expected price decline range of 0.11 to 0.18% per kg of increased BW [35] with seasonal peaks for light steers in November and heavier steers in April.

#### 3.3. Steer Performance

Average ($\pm$SD) on-test weight of steers over the two years was $284.2 \pm 35.9$ kg for steers on the barley diet and $283.3 \pm 36.4$ kg for the triticale diet (data not shown). Average off-test weight of steers for the same period was $392.4 \pm 41.3$ kg for the barley diet and $382.3 \pm 39.3$ kg for the triticale diet.

Regression analysis on observed pen average RFI and diet used categorical variables for barley or triticale diets and quantitative values for the observed RFI for each pen. For DMI, there were no interactions ($p > 0.05$) between diet and the linear or quadratic effects of RFI, no independent linear or quadratic effects ($p > 0.05$) of RFI, and only an effect ($p < 0.01$) of diet (Table 3). The barley silage-based diet DMI, at 5.97 kg steer$^{-1}$ day$^{-1}$, was lower than the 6.42 kg steer$^{-1}$ day$^{-1}$ observed for the triticale silage-based diet (Table 4). This effect of diet on DMI resulted in a difference ($p < 0.01$) in the DMI of barley and triticale silages ($4.45 \pm 0.14$ vs. $4.77 \pm 0.30$ kg steer$^{-1}$ day$^{-1}$) (data not shown).

**Table 3.** *p*-values for Type I effects [1] for diet, observed residual feed intake (RFI), and interactions of diet with RFI.

| Category [2] | Diet | RFI$_{linear}$ | Diet $\times$ RFI$_{linear}$ | RFI$_{quadratic}$ | Diet $\times$ RFI$_{quadratic}$ |
|---|---|---|---|---|---|
| DMI, kg steer$^{-1}$ day$^{-1}$ | <0.01 | 0.31 | 0.71 | 0.37 | 0.17 |
| ADG, kg steer$^{-1}$ day$^{-1}$ | <0.01 | 1.00 | 0.41 | 0.36 | 0.12 |
| Feed efficiency, kg ADG:kg DMI | <0.01 | 0.31 | 0.25 | 0.92 | 0.58 |
| Gross cost, CAD steer$^{-1}$ | <0.01 | 0.31 | 0.71 | 0.34 | 0.18 |
| Gross revenue, CAD steer$^{-1}$ | 0.01 | 0.15 | 0.61 | 0.42 | 0.82 |
| Net revenue, CAD steer$^{-1}$ | <0.01 | 0.02 | 0.40 | 0.86 | 0.53 |
| Breakeven silage value, CAD tonne$^{-1}$ | | | | | |
| At actual DM | 0.14 | 0.03 | 0.36 | 0.91 | 0.87 |
| At 35% DM | <0.01 | 0.03 | 0.48 | 0.86 | 0.60 |
| Methane emissions, CO$_2$e | | | | | |
| kg steer$^{-1}$ day$^{-1}$ | <0.01 | 0.21 | 0.64 | 0.18 | 0.25 |
| kg kg$^{-1}$ DMI | 0.45 | 0.73 | 0.99 | 0.91 | 0.10 |
| Kg kg$^{-1}$ ADG | <0.01 | 0.24 | 0.22 | 0.84 | 0.31 |
| Net revenue, CAD kg$^{-1}$ of CO$_2$e | <0.01 | 0.03 | 0.49 | 0.84 | 0.60 |

[1] Fitted regression equations based on simplest model including all effects with $p < 0.05$ and preceding effects (i.e., if diet had a linear interaction with RFI, then diet and RFI linear effects would be included even if they had $p > 0.05$). [2] DMI, dry matter intake; ADG, average daily gain; DM, dry matter; CO$_2$e, CO$_2$ equivalents.

**Table 4.** Fitted regression equation coefficients for effects of diet, observed residual feed intake (RFI), and interactions of diet with RFI.

| Category [1] | Diet Intercept Coefficients | | | $RFI_{linear}$ Slope | SE [2] |
| | Barley Silage | Triticale Silage | SE [2] | | |
|---|---|---|---|---|---|
| DMI, kg steer$^{-1}$ day$^{-1}$ | 5.97 | 6.42 | 0.15 | | |
| ADG, kg steer$^{-1}$ day$^{-1}$ | 0.97 | 0.88 | 0.03 | | |
| Feed efficiency, kg ADG:kg DMI | 0.16 | 0.14 | 0.0034 | | |
| | | | | | |
| Gross cost, CAD steer$^{-1}$ | 263.19 | 275.61 | 4.11 | | |
| Gross revenue, CAD steer$^{-1}$ | 288.13 | 268.19 | 7.13 | | |
| Net revenue, CAD steer$^{-1}$ | 24.95 | −7.42 | 5.90 | −30.24 | 12.01 |
| Breakeven silage value, CAD tonne$^{-1}$ | | | | | |
| At actual DM | 68.31 | 61.42 | 4.71 | −21.62 | 8.04 |
| At 35% DM | 73.66 | 52.18 | 3.95 | −19.76 | 9.08 |
| Methane emissions, $CO_2$e | | | | | |
| kg steer$^{-1}$ day$^{-1}$ | 3.09 | 3.35 | 0.06 | | |
| kg kg$^{-1}$ DMI | 0.52 | 0.52 | 0.01 | | |
| kg kg$^{-1}$ ADG | 3.23 | 3.81 | 0.08 | | |
| Net revenue, CAD kg$^{-1}$ of $CO_2$e | 7.98 | −2.04 | 1.81 | −9.12 | 3.68 |

[1] DMI, dry matter intake; ADG, average daily gain; DM, dry matter; $CO_2$e, $CO_2$ equivalents. [2] SE, Standard error.

For ADG, there were no interactions ($p > 0.05$) between diet and the linear or quadratic effects of RFI, no independent linear or quadratic effects ($p > 0.05$) of RFI, and only an effect ($p < 0.01$) of diet (Table 3). Steers receiving the barley silage-based diet gained weight faster, at 0.97 kg day$^{-1}$, than steers receiving the triticale silage-based diet, at 0.88 kg day$^{-1}$ (Table 4).

There was year-to-year variation with steer receiving the barley silage-based diet in year 2 having ADG (1.05 kg steer$^{-1}$ day$^{-1}$) that was 117% of the ADG that was observed in year 1 (0.89 kg steer$^{-1}$ day$^{-1}$). Steers receiving the triticale silage-based diet in year 2 had ADG (0.91 kg steer$^{-1}$ day$^{-1}$) that was 105% of the ADG observed in year 1 (0.86 kg steer$^{-1}$ day$^{-1}$). Variation between years can be attributed to the combined impacts of variability in cattle, silage quality, and environment, particularly weather and this necessitated the inclusion of year as a random blocking factor.

Similar to DMI and ADG, for feed efficiency, there were no interactions ($p > 0.05$) between diet and the linear or quadratic effects of RFI, no independent linear or quadratic effects ($p > 0.05$) of RFI, and only an effect ($p < 0.01$) of diet (Table 3). The lower DMI and greater gain of steers receiving the barley silage-based diet resulted in greater feed efficiency, at 0.16 kg ADG–kg DMI, than occurred for the steers receiving the triticale silage-based diet, at 0.14 kg ADG–kg DMI, with their greater DMI and lower ADG (Table 4). The feed conversion ratio (kg DMI–kg ADG; inverse of feed efficiency) was 6.21 and 7.29 kg DMI–kg ADG for the barley silage- and triticale silage-based diets, respectively.

*3.4. Cost, Revenue, Profitability of Steers, and Silage Value*

The gross cost of backgrounding steers was not affected ($p > 0.05$) by any interactions between diet and the linear or quadratic effects, or any independent linear or quadratic effects of cattle efficiency type as assessed by observed pen average RFI (Table 3). There was an effect ($p < 0.01$) of diet on gross cost (Table 3) with steers receiving the barley silage-based diet having a 112-day gross cost of CAD 263 steer$^{-1}$ vs. CAD 276 steer$^{-1}$ for steers on the triticale silage-based diet (Table 4). The average cost for steers on triticale silage was 5% higher than the cost on a barley diet. Feed costs were 62.2% of total costs for steers fed the barley silage-based diet. Silage cost was CAD 80 or 30.5% of the total cost, cDDGS-based supplements CAD 83 or 31.7%, yardage CAD 68 or 26%, bedding CAD 12 or 4.5%, and other costs CAD 20 or 7.3% of the total cost. Similarly, for steers fed the triticale silage-based diet, feed costs were 63.8% of total costs. Silage cost was CAD 86 or 31.3% of the total cost

(CAD 276 steer$^{-1}$), cDDGS-based supplements CAD 90 or 32.5%, yardage CAD 68 or 24.8%, bedding 12 or 4.3%, and other costs CAD 20 or 7.1% of the total cost.

Gross revenue from backgrounding steers was not affected ($p > 0.05$) by any interactions between diet and the linear or quadratic effects, or any independent linear or quadratic effects of observed pen average RFI (Table 3). There was an effect ($p = 0.01$) of diet on gross revenue (Table 3) with steers receiving the barley silage-based diet generating revenue of CAD 288 steer$^{-1}$ vs. CAD 268 steer$^{-1}$ for steers on the triticale silage-based diet (Table 4).

For NR, there were no interactions ($p > 0.05$) between diet and the linear or quadratic effects of RFI, or independent quadratic effects ($p > 0.05$) of RFI (Table 3). However, there was an effect ($p < 0.01$) of diet and a linear effect ($p < 0.05$) of RFI on NR (Table 3). There was a negative return of CAD 7 for the steers fed the triticale silage-based diet, whereas the steers fed the barley silage-based diet had positive return of CAD 25. This is a CAD 32 difference between the two diets. The slope for regression of NR on observed pen average RFI indicates an improvement of CAD 30 in NR for every kg of improvement in RFI (Table 4). With the range in pen average RFI observed in this study (0.86 kg; data not shown) or +1 SD vs. −1 SD (0.50 kg; data not shown), this equates to an improvement in NR of CAD 26 steer$^{-1}$ or CAD 15 steer$^{-1}$, respectively. There was year-to-year variation apparent for NR with steers fed the barley silage-based diet generating CAD 40 steer$^{-1}$ in year 2 compared to CAD 9 steer$^{-1}$ in year 1. Similarly, NR for steers fed the triticale silage-based diet was −CAD 1 steer$^{-1}$ in year 2 compared to −CAD 13 steer$^{-1}$ in year 1.

The values for barley or triticale silages were calculated both at actual DM content and at the industry standard 35% DM content. When calculated at actual DM, there were no interactions ($p > 0.05$) between diet and the linear or quadratic effects of observed pen average RFI, and no effect ($p > 0.05$) of diet or independent quadratic effect of observed pen average RFI (Table 3). There was a linear effect ($p = 0.03$; Table 3) for observed pen average RFI where each kg improvement (reduction) in RFI resulted in an increase in the breakeven value of silage of CAD 22 per tonne (Table 4). With the range in pen average RFI observed in this study (0.86 kg; data not shown) or +1 SD vs. −1 SD (0.50 kg; data not shown), this equates to an improvement in value of silage at actual DM content of CAD 19 tonne$^{-1}$ or CAD 11 tonne$^{-1}$, respectively. The average breakeven value for barley silage at actual DM was CAD 68 tonne$^{-1}$ (Table 4), 132% of the assumed market price (CAD 52 tonne$^{-1}$; Table 2) of barley silage used in determination of NR. For triticale silage, the breakeven value at actual DM (CAD 61 tonne$^{-1}$; Table 4) was 92% of the CAD 67 tonne$^{-1}$ value (Table 2) used in determination of NR.

When the breakeven value of silage was determined at a standardized 35% DM, there was no interaction ($p > 0.05$) between diet and the linear or quadratic, or independent quadratic effects of observed pen average RFI (Table 3). There was an effect of diet ($p < 0.01$) and a linear effect ($p = 0.03$) of observed pen average RFI (Table 3). When standardized to 35% DM, the breakeven value of silage for steers fed the barley silage-based diet, at CAD 74 tonne$^{-1}$, was greater than for triticale silage when fed to steers on the triticale silage-based diet, at CAD 52 tonne$^{-1}$ (Table 4). These breakeven values for barley silage and triticale silage are 130% and 92%, respectively of the silage price suggested through industry standard pricing methods [37]. The linear effect of observed pen average RFI on the breakeven value of silage when standardized to 35% DM was −CAD 20, indicating greater silage value when fed to more efficient cattle types. With the range in pen average RFI observed in this study (0.86 kg; data not shown) or +1 SD vs. −1 SD (0.50 kg; data not shown), this equates to an improvement in value of silage at 35% DM content of CAD 17 tonne$^{-1}$ or CAD 10 tonne$^{-1}$, respectively.

### 3.5. CO$_2$ Equivalent CH$_4$ Emissions

There was no ($p > 0.05$) diet by observed pen average RFI linear or quadratic interactions, or independent linear or quadratic effect of observed pen average RFI for predicted CH$_4$ emissions on kg steer$^{-1}$ day$^{-1}$, kg kg$^{-1}$ DMI, kg kg$^{-1}$ ADG basis (Table 3). There was an effect

($p < 0.01$) of diet on $CH_4$ emissions when expressed on kg steer$^{-1}$ day$^{-1}$, or kg kg$^{-1}$ ADG basis, but not ($p > 0.05$) on a kg kg$^{-1}$ DMI basis. The predicted $CH_4$ emissions, in $CO_2$e units, were lower for steers fed the barley silage-based diet, at 3.09 kg steer$^{-1}$ day$^{-1}$ and 3.23 kg kg$^{-1}$ ADG, than for steers fed the triticale silage-based diet, at 3.35 kg steer$^{-1}$ day$^{-1}$ and 3.81 kg kg$^{-1}$ ADG (Table 4).

The NR per unit of $CH_4$ emissions, on a $CO_2$e basis, was also not affected ($p > 0.05$) by observed pen average RFI linear or quadratic interactions, or independent quadratic effect of observed pen average RFI (Table 3). There was an effect of both diet ($p < 0.01$) and a linear effect ($p = 0.03$) of observed pen average RFI. Steers fed the barley-silage based diet had greater NR, at CAD 8 kg$^{-1}$ $CO_2$e, than steers fed the triticale silage diet, at −CAD 2 kg$^{-1}$ $CO_2$e (Table 4). The slope of −CAD 9 NR kg$^{-1}$ $CO_2$e per kg of observed pen average RFI indicates greater revenue per unit of $CH_4$ emissions with more efficient cattle types. With the range in pen average RFI observed in this study (0.86 kg; data not shown) or +1 SD vs. −1 SD (0.50 kg; data not shown), this equates to an improvement in NR of CAD 8 kg$^{-1}$ $CO_2$e or CAD 5 kg$^{-1}$ $CO_2$e, respectively.

When $CH_4$ emissions were estimated under the hypothetical scenario of steers receiving a un-supplemented (silage-only) diet and factorial effects of supplementation and silage type assessed, there were silage type by supplementation interactions ($p < 0.01$) for emissions on a kg steer$^{-1}$ day$^{-1}$, kg kg$^{-1}$ DMI, and kg kg$^{-1}$ ADG basis (Table 5). Simulations that excluded the cDDGS-based supplement increased ($p < 0.01$) expected emissions on a kg of $CO_2$e steer$^{-1}$ day$^{-1}$ basis for the barley silage diets but decreased ($p < 0.01$) emissions on the same basis when cDDGS-based supplements were removed from the triticale silage diets (Table 5). The lower ($p < 0.01$) kg steer$^{-1}$ day$^{-1}$ emissions estimates from observed performance for barley silage-based vs. triticale silage-based diets were noted earlier and were negated ($p = 0.07$) when comparisons were made between barley silage-only (at 3.19 kg $CO_2$e steer$^{-1}$ day$^{-1}$) and triticale silage-only (at 3.29 kg $CO_2$e steer$^{-1}$ day$^{-1}$) simulated diets (Table 5). Comparisons on a kg kg$^{-1}$ DMI basis only found effects for comparisons of barley silage-based vs. barley silage-only ($p < 0.01$) and triticale silage-based vs. triticale silage-only diets ($p = 0.03$). On a kg of $CO_2$e kg$^{-1}$ ADG, all simple effect methane emissions comparisons differed ($p < 0.01$) and the increase in estimated kg of $CO_2$e kg$^{-1}$ ADG that occurred with switching from barley silage-only to triticale silage-only diets was 224% of the difference estimated previously when switching from barley silage-based to triticale silage-based diets.

**Table 5.** Actual silage-based diet and simulated silage-only methane emissions means.

| Methane Emissions, $CO_2$ Equivalents | Actual Diets (with cDDGS[1] Supplement) | | Simulated Scenario Diets (Silage Only) | | | *p*-Values | | |
|---|---|---|---|---|---|---|---|---|
| | Barley Silage | Triticale Silage | Barley Silage | Triticale Silage | SE[1] | Supplement | Silage | Interaction |
| kg steer$^{-1}$ per day[2] | 3.09 | 3.35 | 3.19 | 3.29 | 0.04 | 0.09 | <0.01 | <0.01 |
| kg kg$^{-1}$ DMI[3] | 0.52 | 0.52 | 0.54 | 0.53 | 0.06 | <0.01 | 0.67 | <0.01 |
| kg kg$^{-1}$ ADG[4] | 3.23 | 3.81 | 4.37 | 5.67 | 0.27 | <0.01 | <0.01 | <0.01 |

[1] cDDGS, corn dried distillers' grains. SE, Standard error. [2] All simple effect mean comparisons are different ($p < 0.01$) except barley silage-only vs. triticale silage-only ($p = 0.07$). [3] DMI, dry matter intake. Barley silage-based diet differs ($p < 0.01$) from barley silage-only, triticale silage-based diet differs ($p = 0.03$) from triticale silage-only, barley silage-based diet does not differ ($p = 0.45$) from the triticale silage-based diet, barley silage-only diet does not differ ($p = 0.12$) from the triticale silage-only diet. [4] ADG, average daily gain. All simple effect comparisons are different ($p < 0.01$).

### 3.6. Beef Diet Ranking: The SERF Approach

The two-sample Hotelling T2 test result ($p = 0.9999$) found no difference between the mean and variances of the observed steer weights from the feeding trial and the mean and variances of the steer weights used in the simulation model (date not shown). Similarly,

there was no difference (*p* = 0.9999) between the mean and variation of the historical and simulated steer prices.

The CDF constructed from simulated steer weight and steer prices showed a preference dominance for treatments. From most to least preferred over most of the range in NR values, the treatment order was efficient steers on barley silage-based diet, inefficient steers on barley silage-based diet, efficient steers on triticale silage-based diets, then inefficient steers on triticale silage-based diets. Treatments converged at extreme expected NR values and there was some crossover between efficient and inefficient steer types on the barley silage-based diet at low expected NR (Figure 1). Use of a negative exponential utility function with the SERF approach (Figure 2) maintained the same treatment ranking with efficient steers on barley silage-based diet being the most risk efficient diet with higher CE across all levels of risk aversion. A risk-neutral producer (ARAC = 0) with efficient steers would require a CAD 32 steer$^{-1}$ premium to choose the triticale silage-based diet over the barley silage-based diet. For the same risk neutral producer, the compensation required to choose triticale silage-based diets over barley silage-based diets increases to CAD 43 steer$^{-1}$ with inefficient steers. A risk neutral producer would value efficient steers on the barley silage-based diet were worth CAD 13 steer$^{-1}$ more than inefficient steers. The value of efficient steers over inefficient steers to a risk neutral producer was CAD 24 steer$^{-1}$ when a triticale silage-based diet is fed. Counter intuitively, there was convergence of treatment preferences as producer risk aversion increased with lower compensation required to encourage more risk adverse producers to switch from barley silage-based diets to triticale silage-based diets or from efficient to inefficient steers. This analysis is based on an RFI difference between efficient and inefficient cattle types of 0.45 kg day$^{-1}$ for steers on the barley silage-based diet and 0.48 kg day$^{-1}$ for steers on the triticale silage-based diet. It does not account for any cost incurred in determining the cattle efficiency type or any agronomic differences in cost of production or yield potential in the production of barley or triticale silages.

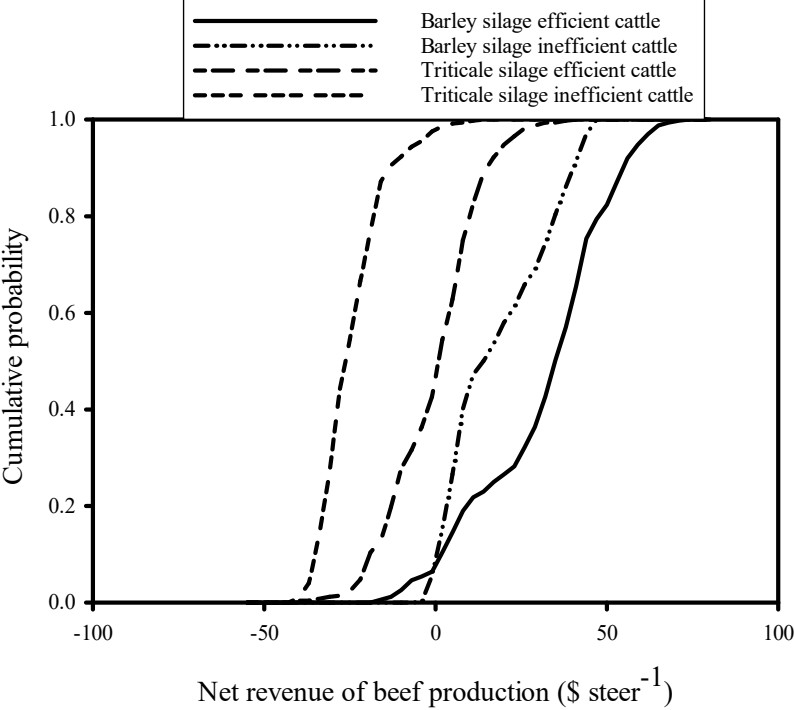

**Figure 1.** The cumulative probability of net revenue of barley and triticale silage-based diets and cattle efficiency type in beef backgrounding steers.

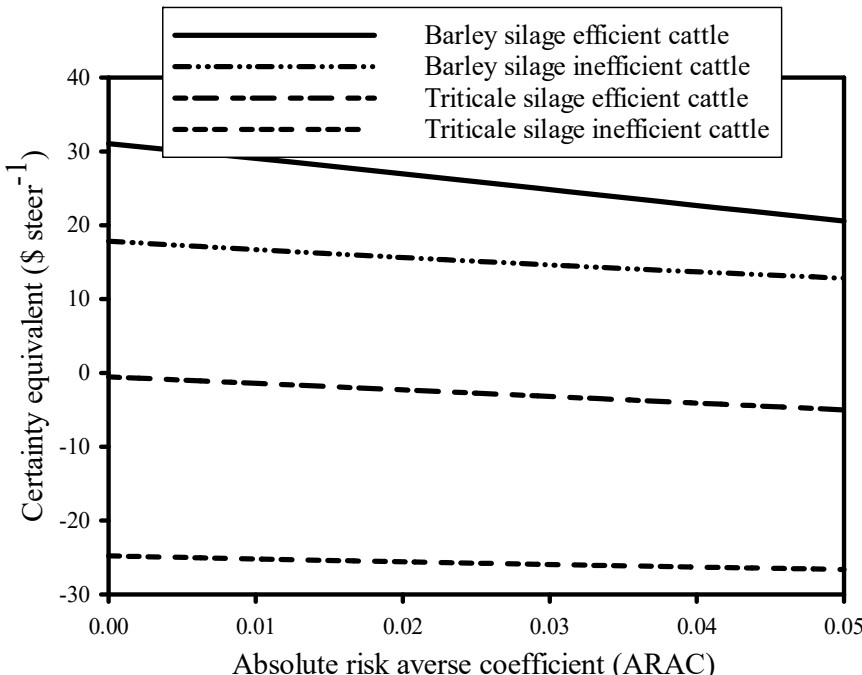

**Figure 2.** Comparison of certainty equivalent of barley and triticale silage-based diets and cattle efficiency type in beef backgrounding steers.

## 4. Discussion

### 4.1. Adequacy of RFI Predictions

Use of predicted RFI in analyzing the effects of RFI is only sensible if there is good agreement between predicted and observed pen average RFI. By definition, the average observed RFI for a contemporary group, all pens of steers in a given year of backgrounding for our study, is zero and was very close to the average predicted RFI in year 1 (0.016 kg) and year 2 (0.014 kg). The stated accuracy for individual steer RFI molecular breeding values averaged 0.35 in year 1 and 0.37 in year 2. This indicates good accuracy. However, good agreement also requires precision. There was a lack of precision with the genomic-based estimates of RFI. The individually predicted RFI values used in assigning cattle to efficiency quartiles and randomizing steers to pens were therefore unsuitable for basing statistical analyses on. This was the reason why our analyses approach changed from a planned 2 (diet) by 4 (cattle efficiency type as defined by RFI) factorial to a 2 (diet) by linear or quadratic cattle efficiency type (defined as observed pen average RFI) factorial.

Conventional RFI testing of beef cattle cannot start before weaning and typically requires an evaluation period in excess of two to three months after a diet adaptation phase [50,51]. This effectively prevents the cattle efficiency type (RFI status) from being measured in advance of the start of the backgrounding period, and in the case of interaction between cattle efficiency type and management (i.e., diet), limits the ability to optimize beef production by matching cattle efficiency types to different management systems. The intention behind our use of genomic testing to allocate weaned calves to cattle efficiency type groups based on predicted RFI was to be able to access and use this information before it could be obtained from conventional assessment methods. Although prediction imprecision with this approach was shown to be ineffective for the current study, future developments in the use of genomic tools to identify difference in cattle efficiency type may be sufficiently improved to allow optimization of beef production by allocation of cattle to various management systems based on cattle efficiency type.

*4.2. Steer Performance and Feed Efficiency of Barley vs. Triticale Silage-Based Diets*

With DMI being a major determinant of production cost and $CH_4$ emissions, ADG affecting gross revenue, and feed efficiency affecting profitability, these aspects of steer performance are important to review in regard to their contributions to the economic and $CH_4$ emissions efficiency of backgrounding steers.

The greater DMI of steers on the triticale silage-based diets than the steers on the barley silage-based diets was not expected. Generally, increased diet energy (digestible, metabolizable, or net energy) content is expected to result in increased voluntary DMI until relatively high diet energy levels are reached [26]. The barley silage had higher energy content than the triticale silage and diets based on both silages received the same cDDGS-based supplement at the same inclusion rate. Even with the increase in diet energy content from the cDDGS-based supplement, neither diet had enough energy that diet energy content was expected to start limiting DMI. Our results contrast with more conventional expectation of lower intake for triticale vs. barley silage-based diets. Our results contrast with more conventional expectation of lower intake for triticale vs. barley silage-based diets due to differences in feed quality with 20% lower intake of triticale vs. barley silage-based diets for weaned heifers observed in a comparable study [52]. McCartney and Vaage [52] also fed the same barley silage and triticale silage to sheep in a digestibility trial and observed DMI by sheep fed the triticale silage to be 58% of the DMI by the sheep fed the barley silage. It is probable that the magnitude of DMI difference between the two trials is due to selectivity difference between cattle and sheep, but it is also possible that providing a greater level of supplementation with the cattle trial mitigated the effect of silage quality on DMI. Their diets used different cultivars of barley and triticale, generally lower quality silage with greater quality difference between silages, and a barley grain-based supplement fed at a lower proportion of total DMI than the cDDGS-based supplement used with diets in our study. Our unexpected observation of greater DMI by steers fed triticale silage-based diets than steers fed barley silage-based diets could be due to differences in silage cultivars, generally higher silage quality with less difference between silage types, higher supplementation levels and use of a cDDGS-based supplement formulated to address an expected metabolizable protein deficiency instead of barley grain-based supplement. In contrast to our observations and the findings of McCartney and Vaage [52], Kennedy et al. [53] compared grass silage, lupin/triticale silage, vetch/barley silage, and blends of grass silage with the other two silages and found no differences in DMI due to silage type.

As a measure of cattle efficiency type that is independent of production [6,7], RFI is moderately heritable and repeatable [51]. Failure to observe a relationship between cattle efficiency type and DMI with our study may be attributed to use of pen-based instead of individual animal observation, allocation of animals to pens based on what was later deemed to be imprecise estimates on RFI, and how the central limit theorem would reduce the relative range in RFI values. A lower range in cattle efficiency type values would make detection of any relationship between RFI and DMI more difficult. Our range in RFI molecular breeding value estimates for individual steers was 0.58 kg in year 1 and 0.50 kg in year 2. For pens of steers, the range in RFI molecular breeding values was reduced by ca. 50% to 0.30 in year 1 and 0.27 in year 2. A similar impact would be expected with the DMI of pens of steers showing a lower range than the DMI of individual steers. Following ca. two generations of selection for divergent RFI, the difference in mean RFI between high and low selection lines was 1.25 kg [7]. The adjusted $R^2$ for our prediction of standardized DMI was 0.49 and compares to a multi-study average of 0.70 for individual cattle [51].

The overall ADG of steers in the study was typical of backgrounding cattle. Observation of greater gain by steers fed the barley silage-based diet than steers fed the triticale silage-based diet was as expected. This observation is comparable to that of McCartney and Vaage [52] who found ADG for heifers was greater with barley silage-based diets (0.65 kg day$^{-1}$) than with triticale silage-based diets (0.49 kg day$^{-1}$). The smaller difference between silage-based diets in the current study vs. the study by McCartney and Vaage [52]

can be attributed to the greater intake of the triticale silage-based diet in the current study partially compensating for the lower quality of the silage. Kennedy et al. [53] did not observe any differences in the ADG of steers fed lupin/triticale or vetch/barley silages, with or without inclusion of grass silage. The lack of effect of cattle efficiency type, as based on the observed pen average RFI, is entirely attributable to the use of pen average ADG in the determination of RFI and is why the *p*-value for the effect of RFI on ADG is 1.00.

With diet being the only factor affecting the DMI and ADG of steers, it is not surprising that the feed efficiency of steers was also affected only by diet. Characterization of cattle efficiency type using observed pen average RFI meant that cattle efficiency type was already adjusted for differences in in cattle size, ADG, and composition of gain before comparing against feed efficiency. The lower DMI and greater ADG of steers fed the barley silage-based diets combined to result in better feed efficiency than the steers fed the triticale silage diets. Although greater DMI with the triticale silage-based diet was unexpected, greater ADG and better feed efficiency with the barley silage-based diet were expected given the better quality of the barley silage and the resulting diet. McCartney and Vaage [52] presented feed conversion instead of the more appropriate feed efficiency and found triticale silage-based diets, at 109% of barley silage-based diet, not to differ from barley silage-based diets. The reciprocals of the feed conversion values reported by McCartney and Vaage [52] indicate feed efficiency values that ca. 60% poorer than observed in the current study. Reasons for better feed efficiency than reported previously include a higher inclusion level of a cDDGS-based supplement formulated to address an expected metabolizable protein deficiency and support efficient lean gain, but also supplying more energy than a barley grain-based supplement, and use of steers with more lean growth potential than heifers. Most of these same reasons also apply when comparing the current study findings to the observations of Lopez-Campos et al. [54]. The feed efficiency observed in the current study approached levels more commonly expected for feedlot cattle receiving a high-grain finishing diet [8]. Explanations for this high level of feed efficiency on a forage-based diet include the high quality of the silages used and the increase in diet energy level that resulted from use of a cDDGS-based supplement to increase metabolizable protein supply while still limiting the risk of ruminal acidosis.

*4.3. Cost, Revenue, Profitability of Steers, and Silage Value*

Overall, feed cost was 62.2% of the total cost for barley silage/corn supplements and 63.8% for triticale silage/corn supplements. Greater feed cost for the steers fed the triticale silage-based diets are attributed to the greater DMI of these steers as the unit cost of silage and supplement was the same for both diets. Randomization of steers to pens to equalize initial weight and weight distribution resulted in equal steer purchase prices. The remainder production costs for backgrounding, including bedding, yardage, marketing and transport costs, were also equal across treatments. A survey of beef producers in Alberta showed that feeding and management through the winter comprise up to two-thirds of the total cost of primary beef production [3]. A guideline developed by Manitoba Agriculture for backgrounding cattle production [55] has shown that feed costs comprise about 61% of the operating costs of backgrounding commercial cattle production. Generally, feed costs amount to 61% of total costs for a typical backgrounding operation [56]. This supports our observed cost distribution as being representative of industry practice.

The greater gross revenue for steers fed the barley silage-based diet is due to the greater ADG for these steers in response to the greater quality of the barley silage vs. the triticale silage. Large price discounts associated with excess size for slaughter cattle are not common with backgrounding cattle. Increased steer weight at the end of the backgrounding period in the current study increased gross revenue.

Lower feed costs, similar to other production costs, and greater gross revenue for steers fed the barley silage-based diet combined to result in greater net revenue than observed with steers fed the triticale silage-based diet. Improved performance of cattle resulting from feeding better quality diets of similar cost should improve economic returns

from cattle production. Negative net returns for the steers fed the triticale silage-based diets are not unexpected. Profit margin for a backgrounding operation in western Canada is about CAD 30 steer$^{-1}$ not including risk management cost [56]. Profit margin could go to negative CAD 24 steer$^{-1}$ if risk management cost for backgrounding operation is included. Various industry-based assessments of backgrounding profitability indicate negative returns from backgrounding. Return on investment and return on asset for a typical backgrounding operation in Manitoba were estimated to be negative 2.2% and 7.7%, respectively [55]. Backgrounding systems decisions have implications for other beef production segments and are often focused on minimizing costs. Our finding of marginal or positive returns in this study suggest an opportunity for improved economic performance from backgrounding as a result of increased attention to diet quality and animal performance.

Despite the lack of effect on DMI or ADG, more efficient cattle types, based on observed pen average RFI, were shown to be more profitable. With RFI identifying cattle with lower feed intake, given weight, ADG, and composition of gain, more efficient cattle types are expected be more profitable. Post-trial pen average RFI did confirm economic value for more efficient cattle types that only requires improvement in early prediction for beef producers to be able to benefit from. Dividing the pens of steers into efficient (RFI < 0) and inefficient (RFI > 0) groups would have resulted in the efficient pens of steers having a CAD 12 NR advantage. This compares to the CAD 32 observed benefit to NR from difference in diet quality which was 2.68 times the benefit from feed efficiency.

Silages are rarely traded feeds making price discovery difficult. A common approach to price determination is base silage value on a conversion from alternate crop uses. An alternative for value determination would be to calculate silage value based on operation breakeven and known costs for all other activities. Assuming all other prices and costs are known except the price and cost of silage, a breakeven value (NR = 0) was calculated for the barley silage and triticale silage as used in the current study. This approach integrates management consideration into silage value and would reflect a market reality that differences in management can affect silage value. Silage costs above these breakeven values would not be acceptable and prices below these values would represent increased opportunity for profit. Our study findings indicated the higher quality of barley silage supports a higher valuation than the currently assumed industry price. Alternatively, triticale silage valuation was below the currently assumed industry price. Silage value also increased with more efficient cattle types indicating the returns expected for each silage would increase when the silage was used more effectively by the steers. Our evaluation does not address agronomic considerations in silage production in relation to silage yields or silage cost of production. It is possible that agronomic considerations could alter silage preference for backgrounding relative to just steer economic based preferences.

*4.4. CO$_2$ Equivalent CH$_4$ Emissions*

An assessment of emissions is an item of increasing importance with growing awareness and attention to the role of greenhouse gas emissions in climate change. The CH$_4$ from all enteric fermentation represents just over 3% of all Canadian greenhouse gas emissions [57]. Relative to cow-calf and feedlot segments of beef production, the smaller size of backgrounding cattle and shorter duration of the backgrounding phase result in a reduced contribution to these emissions.

The observation of lower estimated CH$_4$ emissions for steers fed the barley silage-based diet on a per steer day$^{-1}$ basis but not on a per kg DMI basis indicates the main driver of difference in estimated CH$_4$ emissions was the greater DMI of steers fed the triticale silage-based diet. Comparably, greater ADG for the steers fed the barley silage-based diet and similar CH$_4$ emissions per steer day$^{-1}$ resulted lower emissions per unit ADG than for steers fed the triticale silage-based diet. The impact of diet and cattle efficiency type on the NR per unit emissions indicate that both the barley silage-based diet and the lower observed pen average RFI cattle have a measure of revenue that could be used to offset

economic pressure to reduce $CH_4$ emissions. This was not the case for steers fed the triticale silage-based diet, or for pens of steers with lower observed pen average RFI.

For the hypothetical situation of feeding silage-only diets (eliminating cDDGS-based supplement), in all cases of $CH_4$ emissions per steer day$^{-1}$, per unit DMI, or per unit ADG, there were interactions between silage type and the inclusion or removal of the cDDGS-based supplement although the nature of the interactions differed.

Eliminating cDDGS-based supplement substantially reduced expected ADG, more so for triticale silage than barley silage, and resulted in much higher predicted emissions rates. This was based on energy limited estimates of ADG and not the more restrictive protein limited estimates of ADG. Supplementation to address nutrient imbalances was predicted to substantially improve the efficiency of beef production on a $CH_4$ emissions basis. Impact of cattle efficiency type was not considered in these comparisons as, apart from using observed performance, the NASEM [26] models have no mechanism to account for differences in cattle efficiency type as related to RFI.

### 4.5. Beef Diet Ranking: The SERF Approach

Economic analyses expanded beyond focusing just on point estimates, such as treatment means for NR. A risk ranking procedure that does not rely on point estimates or summary statistics is and included the CDF and SERF analysis. A prerequisite for these additional analyses is that the Hotelling $T^2$ tests for actual and simulated steer weights and feeder prices were not significant. This was the case and SIMETAR [43,45] was an appropriate method tool to be used to construct CDF from simulated steer weights and feeder prices. For efficient steers fed barley silage-based diets to be completely dominant alternatives for beef returns would require the efficient steers fed barley silage-based diets to lie on the right-hand side of the CDF curves throughout the entire range of alternative return distributions (Figure 1). It is apparent that, through most of the CDF range, use of the efficient steers on higher quality barley silage-based diet would be the preferred option for improving net revenues. The NR advantage of efficient steers over inefficient steers fed the barley silage-based diet is generally lower, even reversing at low NR expectations, than the NR advantages observed when the triticale silage-based diet was fed, suggesting the economic premium for efficient cattle type declines with higher quality diets.

Since each individual producer may have different risk aversion and differences in risk aversion can affect treatment preferences, additional analyses beyond CDF are necessary to assess the impact of risk aversion on producer preferences. The alternative method of SERF was used to assess the preferred treatment as it is an analytical technique of greater preference because it incorporates risk preference of individual producers [47,48]. Similar stochastic variables have been used in previous beef studies [28,58–62]. Khakbazan et al. [30,31] reported a similar SERF analytical technique for ranking different beef management systems.

Figure 2 presents the SERF approach using a negative exponential utility function. This method identifies utility efficient alternatives for a range of risk preferences by ranking alternatives in terms of CE [30,31]. The SERF values results are consistent with other economic findings, strengthening support for selection of efficient steers and use of barley silage-based diets over inefficient steers or triticale silage-based diets by both risk neutral and risk averse beef backgrounders. Difficulty in determining cattle efficiency type in advance of backgrounding, as observed with the results of the current study, makes adoption of differential pricing based on cattle efficiency type challenging. Although strong risk aversion due the typical low margins from backgrounding cattle and converging treatments with increasing risk aversion will reduce preference differences between use of barley silage-based diets and triticale silage-based diets for backgrounding beef steers, there would need to be substantial agronomic considerations for triticale to become preferred over barley for production of silage-based diets for backgrounding steers.

## 5. Conclusions

This study used data collected over two years, 2016/2017 and 2017/2018, and showed that the profitability of beef backgrounding was affected independently by cattle efficiency type and diet quality but not with their interactions, with higher quality barley silage also lowering $CO_2$e emissions. Assessment of RFI prediction adequacy found no relationship between observed RFI and predicted pen average RFI. The barley silage treatments typically provided greater ADG and better feed efficiency, higher NR, and lower $CO_2$e emissions compared to triticale silage treatments. Regression results also showed that NR, but not ADG, was affected by cattle efficiency type. The risk simulation analysis mirrors the findings of the experimental NR results, as producer preferences should be for selection of efficient steers over inefficient steers and barley silage-based diets over triticale silage-based treatments. The difference in certainty equivalent (averaged CAD ~30 steer$^{-1}$) of efficient steers on barley silage and inefficient steers on barley silage or efficient or inefficient steers on triticale silage supports a beef backgrounding producer focus on diet quality and cattle efficiency type. However, inefficient steers on barley silage still generated a higher certainty equivalent per steer than efficient or inefficient steers on triticale silage. The observed benefit to NR from difference in diet quality was 2.68 times the benefit from cattle efficiency type measured by observed pen average RFI. This study did not address potential agronomic differences, including yield, which could provide nuance to forage choice. To increase understanding and improve assessment of silage diets in relation to profitability, additional models that address agronomic difference between production of barley and triticale silages should be built.

**Author Contributions:** M.K. and H.C.B. wrote the manuscript, with input from J.J.C., V.S.B., J.A.B., C.L. and C.E.-D. H.C.B. contributed to the research design, project administration, and data collection. J.H. contributed to the processing of the data. All authors contributed to the revision of the manuscript. M.K. is responsible for the manuscript as a whole. All authors have read and agreed to the published version of the manuscript.

**Funding:** Agriculture and Agri-Food Canada provided funding for this study (Project J-001336).

**Institutional Review Board Statement:** All dietary treatments and experimental procedures related to animal use and care were approved on 18 April 2016 by the Lacombe Research and Development Centre Animal Care Committee (LRC Study Plan No. 201602) prior to any animal-related research activity with animal care in accordance with the guidelines established by the Canadian Council on Animal Care [32]. An amendment to allow additional sampling was approved on 18 January 2017 with interim and final animal use reports accepted on 18 April 2017 and 16 April 2018. The study complies with the ARRIVE guidelines.

**Informed Consent Statement:** Not Applicable.

**Data Availability Statement:** The datasets for the current study are available from the corresponding authors on reasonable request.

**Acknowledgments:** The financial support of Agriculture and Agri-Food Canada is gratefully acknowledged. Technical assistance provided by Rebecca Xie is greatly appreciated. The authors thank Michael Vinsky for coordination of SNP genotyping on the animals. We also thank all the researchers, technicians and support staff at the Agriculture and Agri-Food Canada's Lacombe Research and Development Centre in Lacombe, AB.

**Conflicts of Interest:** Agriculture and Agri-Food Canada provided funding for this study. Apart from this, no competing interests need to be declared.

## Abbreviations

| | |
|---|---|
| ADG | average daily gain; |
| ARAC | absolute risk aversion coefficient; |
| BW | body weight; |
| cDDGS | corn (*Zea mais*) dried distillers' grains; |
| CDF | probability distribution function; |
| CE | certainty equivalent; |
| DMI | dry matter intake; |
| gEPD | genomic expected progeny differences; |
| NASEM | National Academies of Sciences, Engineering, and Medicine; |
| NR | net revenue; |
| RFI | residual feed intake; |
| SERF | stochastic efficiency with respect to a function |

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
