# Peer review of "Effects of Silage-Based Diets and Cattle Efficiency Type on Performance, Profitability, and Predicted CH4 Emission of Backgrounding Steers"

_agriculture, doi:10.3390/agriculture12020277_

Round 1

Reviewer 1 Report

Agriculture ID 1571969

In the manuscript the effects of the silages (barley and triticale) inclusion in the diets of steers on performance, economy advantages and the prediction of CO2 emission.

Please, check the format of the manuscript and correct it according to the template’ journal.

First of all, the manuscript lacks the ethical clearance authorization of experiment.

I suggest to change the title in: Effects of Silage Based Diets and Cattle Efficiency Type on Performance, Profitability, and Predicted CH4 Emission in Steers

The paper it’s interesting but the information are too much for a single paper and it’s difficult to understand all the showed data. I suggest to divide in two article (section A and section B) the first one for the animals performance and the prediction of methane production and the second one with the economic evaluation.

Introduction:

-Please add in the introduction an hypothesis for you study.

-Line 75 add a reference in order to support this sentence.

-Why do you have chosen to change the silages in the diets? Add information concerning the choice of the silages and the performance obtained in other studies present in literature.

Mat & Met: please add live weight and age (mean +/- stardand deviation at time 0)

Lines 179-190: please move this period below, the statistical analysis could be related only to the model used for the data analysis.

I suggest to add also the formula of the statistical analyses.
In materials and methods must to be added:

  • The diets composition
  • The chemical composition and the energy value of the diets utilized in to the experiment and, also, the chemical composition of the silages. Have you performed these analyses? It’s really important in order to discuss the differences in steers performance.

Add in the footnotes of the table the SE.

Discussion: I suggest to divide the discussion as the results section according to the different results (i.e. Adequacy of RFI predictions, Economic analyses data, Steer performance etc.) in order to better understand it.

Line 457-462: please revise this period, it’s very confused.

Please, revise the discussion concerning the animals performance according to the diets characteristics and composition.

Conclusions: the conclusion section must be rewrite and reduced by focusing on the results and the possible explanation of the obtained results.

Author Response

Thank you very much for your review. Please see attached my point-by-point response to Reviewer 1

Reviewer 2 Report

The paper which focuses on beef steers categorised using residual feed intake and its interaction effect on diet quality was well laid-out. The results and discussion integrate the data as well as the implications of the study. 

However, the manuscript appears lengthy and the authors will do well to consider reducing the overall length where possible.

Specific comments are provided below:

L68-71: Sentences need to be re-casted. A suggestion is provided.

The responsiveness of cattle to different quality diets is relatively well understood [26] allowing joint consideration of yield and quality with crop selection models [27]. There exist some research evidence that different crop species such as Canmore barley and Bunker triticale, although differ substantially in quality, but they have comparable economic value [27].

L79: are may be more…delete “are”

L118: “ultrasound backfat thickness measures, were collected throughout the study”…How often was this measured? Monthly? Weekly or at the end of the trial In each year. Please specify.

Economic analysis: I observed that in calculating feed cost, Maize grain DDGS was not included in the NR model. This may not be important only if it is established that the consumption of cDDGS was uniform across the treatments (efficiency type and diet type). With barley and triticale differing in quality, will cDDGS intake not likely vary? Authors may need to provide justification for this.

L266-267: It is not easy to link the trial weight range (e.g 136-181 kg) with the month as depicted in Table 2. This information should be added for easy reading.

L286-287: Could the animals consuming 5.97 kg/day of barley silage for example have made up with higher intake of cDDGS as compared to steers consuming 6.42 kg/day of triticale silage. How does this possibility affect overall RFI?

L307: Sentence correction. …and this necessitated the inclusion of year as a random blocking factor.

L309: comma after efficiency.        

L314: feed conversion ratio?

L363-375: remove italics

L472: delete “of cattle”

L480: in the use of genomic tools

L489: DMI of steers

L496: recast sentence please.

L623-646. I think the discussion section was too lengthy. Information on silage pricing and the alternative options can be shortened especially because some background information had been earlier provided in the material ad method section.

Author Response

Thank you very much for your review. Please see attached my point-by-point response to Reviewer 2

Round 2

Reviewer 1 Report

the paper could be accept in the current form.